# Cost of Illness in Young Children: A Prospective Birth Cohort Study

**DOI:** 10.3390/children8030173

**Published:** 2021-02-24

**Authors:** Sarah Kristine Nørgaard, Nadja Hawwa Vissing, Bo Lund Chawes, Jakob Stokholm, Klaus Bønnelykke, Hans Bisgaard

**Affiliations:** Copenhagen Prospective Studies on Asthma in Childhood (COPSAC), Herlev and Gentofte Hospital, University of Copenhagen, 2820 Gentofte, Denmark; sarahkristinepedersen@gmail.com (S.K.N.); nadja.hawwa.vissing@regionh.dk (N.H.V.); chawes@copsac.com (B.L.C.); stokholm@copsac.com (J.S.); kb@copsac.com (K.B.)

**Keywords:** cesarean delivery, costs of illness, childhood

## Abstract

Childhood illness is extremely common and imposes a considerable economic burden on society. We aimed to quantify the overall economic burden of childhood illness in the first three years of life and the impact of environmental risk factors. The study is based on the prospective, clinical mother–child cohort Copenhagen Prospective Studies on Asthma in Childhood (COPSAC2010) of 700 children with embedded randomized trials of fish-oil and vitamin D supplementations during pregnancy. First, descriptive analyses were performed on the total costs of illness, defined as both the direct costs (hospitalizations, outpatient visits, visit to the practitioner) and the indirect costs (lost earnings) collected from the Danish National Health Registries. Thereafter, linear regression analyses on log-transformed costs were used to investigate environmental determinants of the costs of illness. The median standardized total cost of illness at age 0–3 years among the 559 children eligible for analyses was EUR 14,061 (IQR 9751–19,662). The exposures associated with reduced costs were fish-oil supplementation during pregnancy (adjusted geometric mean ratio (GMR) 0.89 (0.80; 0.98), *p* = 0.02), gestational age in weeks (aGMR = 0.93 (0.91; 0.96), *p* < 0.0001), and birth weight per 100 g (aGMR 0.98 (0.97; 0.99), *p* = 0.0003), while cesarean delivery was associated with higher costs (aGMR = 1.30 (1.15; 1.47), *p* < 0.0001). In conclusion, common childhood illnesses are associated with significant health-related costs, which can potentially be reduced by targeting perinatal risk factors, including maternal diet during pregnancy, cesarean delivery, preterm birth and low birth weight.

## 1. Introduction

Childhood illness is very common in the general population, particularly in the first three years of life [1]. The majority of illnesses are simple infections such as respiratory, gastrointestinal or feverish episodes of short duration. Other common childhood illnesses include allergies, skin problems, eye conditions, neurological issues and gastrointestinal conditions. The majority of these illnesses are mild and easily treated, often without or with very brief healthcare contact, but due to their high frequency, they constitute a sizeable economic burden to society due to healthcare utilization and parents’ work absenteeism [1]. Nevertheless, knowledge on the economic costs of childhood illness is sparse despite the fact that the percentage of Gross Domestic Product used on healthcare is increasing and puts pressure on the welfare systems of industrialized countries [2,3]). Most studies on the costs of childhood illnesses focus on the costs of a single illness, e.g., asthma [4,5,6], or a group of related illnesses such as lower respiratory tract infections [7]. So far, no studies have dealt with the total costs of common childhood illnesses, including nonsevere self-limiting infections, which are not easily studied in larger populations, since they often do not require visits to medical facilities, but can be treated at home.

There are a few known modifiable risk factors that increase childhood illness in general, which can be targeted to reduce costs. Cesarean delivery is one such risk factor, which has increased due to changes in obstetrical practices, medical risk profiles and increasing “maternal request”. Children delivered by cesarean section have an increased risk of a range of immune-related diseases such as asthma, inflammatory bowel disease, leukaemia, and immune deficiencies [8], which might be reflected in the overall health-related costs, tracking from birth into early childhood.

We have previously shown in a randomized controlled trial (RCT) that maternal fish-oil supplementation in pregnancy reduces the risk of asthma and lower respiratory tract infections in childhood [8,9]. Likewise, we showed that vitamin D supplementation in pregnancy may also have clinically important protective effects on the same type of illnesses [10], which are some of the most common chronic childhood illnesses [11] and therefore a substantial contributor to health-related costs in childhood. These pregnancy dietary interventions, cesarean section and possibly other modifiable environmental risk factors therefore have the potential for reducing the health-related costs of childhood illness.

The aim of this study was to estimate the overall economic burden on society caused by childhood illness the first three years of life by using data from a prospective, clinical population-based mother–child cohort study. Secondly, to investigate whether a range of early life environmental risk factors, including the effects of dietary supplements with vitamin D and fish-oil in pregnancy are related to the costs of illness in early childhood in order to identify targets to reduce costs of childhood illness.

## 2. Materials and Methods

### 2.1. Study Population

COpenhagen Prospective Study on Asthma in Childhood (COPSAC_2010_) is a longitudinal, population-based mother–child cohort study of 700 Danish children born from 2009 to 2011. The mothers were included during pregnancy and the children were followed closely from birth with scheduled clinical visits as well as acute visits for any acute airway and/or skin symptoms. The cohort study design and baseline characteristics have previously been thoroughly described [12]. Data validation and quality control followed the guidelines for good clinical practice.

### 2.2. Illness

Child illness was monitored by health interviews with the parents during visits at the COPSAC clinic supported by daily diary cards filled prospectively from birth by parents monitoring the child’s symptoms of illness between clinic visits. The diary cards were reviewed with the family by the COPSAC pediatricians at each visit to validate symptom definitions and entries. For the first three years of life the children were seen at age 1 week, 1 month, 3 months, 6 months and six-monthly hereafter. Prior to the health interview sessions, data were retrieved from the regional hospital register and The Danish National Prescription Register [10,13] on hospital admissions, outpatient contacts and prescribed medication to be confirmed by the parents at every visit.

### 2.3. Parent Work Absenteeism

In the diary cards, parents also specifically registered daycare absenteeism due to illness. The number of days home from daycare causing parent absenteeism due to illness were estimated based on this diary registration. Only weekdays (Monday–Friday) were included in the estimation.

### 2.4. Estimation of Costs

The costs of illness were investigated as “direct costs”, “observed total costs” and “standardized total costs”: “Direct costs” were defined as the direct medical expenditures associated with contact with the healthcare system, including payments for hospital outpatient services, hospital inpatient stays, emergency department visits, physician and facility payments. Data were retrieved from the Danish National Health Registers (The Danish National Health Service Register [14] and the Danish National Patient Register [15], which encompass information on all contacts a citizen has with the healthcare system with subsequent fixed reimbursements to the healthcare provider. The total reimbursement related to an episode of illness was used as a proxy for the direct costs of the medical event.

“Direct costs” were grouped according to the type of healthcare service: “physician costs”, “hospital admission costs” and “hospital outpatient costs". “Physician costs” include costs related to visits to private practitioners, including general practitioners and specialized doctors. “Hospital costs” include the costs of hospital admissions estimated by the reimbursement received by the hospital after treatment has ended. “Outpatient costs” include the costs of outpatient hospital visits estimated as the reimbursement received by the hospital after each visit. The “indirect costs” include lost earnings due to parent absenteeism when their child is ill. The indirect costs were estimated as both “observed indirect costs”, based on the annual household income adjusted for the number of parents in the household, and as “standardized indirect costs” based on a standardized salary [16]. The “total cost” per child was estimated as the sum of all direct and indirect costs of illness during the child’s first three years of life with “observed total costs” being based on observed household income and “standardized total costs” being based on a standardized salary. All costs were stated in EUR 2017.

### 2.5. Pregnancy Interventions

Two double-blind RCTs were conducted in the COPSAC_2010_ cohort: (1) Mothers were assigned to a daily supplement of either 2.4 g n-3 LCPUFA (fish-oil) or control (olive oil) [9]. (2) Mothers were assigned to a daily supplement of either high dose vitamin D3 (2800 IU) or standard dose (400 IU) [9,10]. Both supplements were given in a 2 × 2 factorial design from week 24 of pregnancy to 1 week after birth.

### 2.6. Environmental Risk Factors

Highly detailed risk factor assessment is a key feature of the COPSAC_2010_ study, see http://copsac.com/available-data/ (accessed 20 February 2020). This includes information on a wide range of environmental and constitutional factors collected prospectively [12]. On the basis of the literature [17,18,19,20,21,22,23,24,25], 18 environmental risk factors were chosen a priori for suspected relevance for incidence of childhood illness: maternal smoking during 3rd trimester of pregnancy (yes/no); maternal age at birth (years); maternal allergic disease (asthma, eczema or rhinitis); maternal prepregnancy BMI; preeclampsia (yes/no); cesarean delivery (yes/no); sex (male vs. female); prematurity (<37 weeks of gestation) (yes/no); gestational age (GA) (weeks); birth weight (100 g); season of birth (spring, summer, fall or winter); older siblings at birth (yes/no); living with cats at home at birth (yes/no); living with dogs at home at birth (yes/no); duration of solely breastfeeding (months); social circumstances (see Appendix B for definition); day care attendance (age at start in years) and type of day care (nursery vs. private daycare).

### 2.7. Statistics

Descriptive statistics of the costs of child illness and the number of days with parental work absenteeism due to child illness were performed. The incidences of disease during the first three years of life were estimated by ICD-10 group. Differences in baseline characteristics between groups were compared with Chi-squared test for categorical variables and Student’s t-test for continuous variables.

The effects of the pregnancy interventions with fish-oil and high-dose vitamin D on the costs of child illness were investigated with linear regression models on log-transformed costs of illness (standardized total costs, observed total costs, direct costs), estimated as crude Geometric Mean Ratios (GMR). The effects of the interventions on number of days absent from daycare were investigated with Poisson regression models estimated as Odds Ratios (OR).

Thereafter, linear regression models on the log-transformed costs adjusted for the fish-oil intervention were used to investigate associations between environmental factors and costs of illness for each environmental factor. Poisson models adjusted for the fish-oil intervention were used for associations between environmental factors and daycare absenteeism. A multivariate model excluding collinear factors was fitted for all factors, which were significantly associated with more than one type of costs. Additionally, the actual costs were calculated based on the estimates and geometric mean (GM). For parental work absenteeism, the costs were standardized based on average daily salaries in three countries: Denmark, the United Kingdom and the USA.

We performed a subanalysis excluding children of mothers who received active fish-oil supplementation, previously shown to be significantly associated with childhood illness [26], for generalizability to a nontreated population. We also performed subanalyses elaborating the association between cesarean delivery and costs of illness, stratifying by mode of delivery and age in years, and adjusted for GA and birth weight. We performed sensitivity analyses investigating only: (1) children born after GA 37 weeks, (2) children without asthmatic symptoms before age 3, and (3) the costs of hospitalizations. Additionally, a linear regression model was used to investigate the association between log-transformed costs and type of cesarean section (planned vs. acute) in the subpopulation born by cesarean delivery.

A significance level of 0.05 and 95% confidence intervals were used unless otherwise stated. Missing data were treated as missing observations. All statistics were performed with R (version 3.5.1 (2 July 2018)). All results are stated in EUR 2017.

## 3. Results

### 3.1. Baseline Characteristics

Of the 700 children included in the COPSAC_2010_ mother–child cohort, we excluded children who did not complete the first 3 years of the study, children with less than 90% of the daily diary registrations and children not registered in the National Danish Register, leaving 559 (80%) eligible children for analysis (Appendix A). The baseline characteristics of the participants are summarized in Table 1. The children in the study population were less often born by cesarean delivery and had fewer dogs at home at the time of birth compared to the children excluded (Appendix A).

Figure 1 shows the burden of child illness experienced during the first three years of life according to the ICD-10 classification [27]. A total of 311 (56%) of the children were hospitalized during the first three years of life, while 231 (41%) of the children were hospitalized after the age of 30 days.

### 3.2. Costs of Illness

Figure 2 shows the distribution of the different types of costs and Appendix A displays the descriptive statistics of the costs and days absent from daycare during the child’s first three years of life. The median crude overall costs of childhood illness were EUR 10,932 per child throughout the first three years of life, with a median direct cost of EUR 4225. When indirect costs due to parental work absenteeism were estimated using standardized salaries, the total costs were EUR 13,290. In the subgroup of children (*n* = 276) whose mothers did not receive fish-oil supplementation the costs were slightly higher: observed total costs, EUR 11,987; direct costs, EUR 4372; and standardized total costs, EUR 14,061, respectively. Excluding children (*n* = 158) whose mothers received high-dose vitamin D supplementation resulted in similar estimates: observed total costs, EUR 11,469; direct costs, EUR 4116; and standardized total costs: EUR 13,553.

The crude median number of days absent from daycare due to illness was 24 days (IQR 15–37). When accounting for time enrolled in daycare, the median number of days was 12 days per year in day care (IQR 7–17). In the subgroup of children whose mothers did not receive fish-oil supplementation the median number of days absent was 26 days (IQR 16–39). Excluding children whose mothers received high-dose vitamin D supplementation resulted in a similar estimate of 26 days (IQR 14–41).

### 3.3. Pregnancy Nutritional Interventions

Fish-oil supplementation in pregnancy was associated with lower total costs during the first three years of life (standardized total costs: GMR = 0.89 95% CI = (0.80; 0.98), *p* = 0.021) and showed a trend of association with lower direct costs (GMR = 0.88 (0.76; 1.02), *p* = 0.09) and number of days absent from daycare (OR = 0.90 (0.80; 1.01), *p* = 0.07). There were no associations between high-dose vitamin D supplementation in pregnancy and costs or number of days absent from daycare (*p*-values >0.15). The baseline characteristics of the four intervention groups are summarized in Appendix A.

### 3.4. Environmental Risk Factors

Table 2 shows the associations between environmental risk factors and costs and days absent from daycare (adjusted for fish-oil supplementation in pregnancy) along with an estimation of the related costs. Environmental factors significantly associated with standardized total costs, observed total costs and direct costs of illness were cesarean delivery, sex, GA, prematurity, and birth weight. Cesarean delivery was also significantly associated with an increased number of days absent from daycare. Appendix A shows the corresponding costs of parental absenteeism based on average daily salaries in Denmark, the United Kingdom and the USA. Higher age at introduction to daycare was associated with lower total costs and attending a nursery compared to a private daycare was associated with higher costs. The direct costs were not associated with the number of days absent from daycare. Limiting the study population to the children whose mothers did not receive fish-oil supplementation during pregnancy did not alter the results (Appendix A).

We performed a multivariate analysis including the variables: cesarean delivery, GA, birth weight, sex, and fish-oil supplementation on the costs of illness (standardized total costs, observed total costs and direct costs). In this model, cesarean delivery remained significant for all types of costs, e.g., standardized total costs: aGMR = 1.24 (1.09; 1.40), *p* = 0.001, whereas sex remained significantly associated only with the total costs: standardized total costs: aGMR = 1.15 (1.04; 1.27), *p* = 0.006. Due to the strong collinearity between GA and birth weight (r = 0.65), each of these variables were excluded when evaluating the other. Both remained significant for all three types of costs, e.g., aGMR estimates for standardized total cost were 0.94 (0.92; 0.97) and 0.98 (0.97; 0.99) per week GA and per 100 g birth weight, respectively. There were no significant associations with birth weight when adjusting for GA (data not shown).

### 3.5. Cesarean Delivery

A total of 110 (20%) children were born by cesarean delivery, of which 49 (45%) were planned procedures. The association between cesarean delivery and the costs of illness remained significant after adjustment for GA, birth weight and fish-oil supplementation: standardized total costs: aGMR = 1.24 (1.10; 1.41), *p* < 0.001; and direct costs: aGMR = 1.25, (1.04; 1.50), *p* = 0.017. Excluding costs of hospitalizations (standardized total costs: aGMR = 1.17 (1.04; 1.33), *p* = 0.011; and direct costs: aGMR = 1.17, (1.01; 1.35), *p* = 0.037) or excluding children born before GA 37 weeks from the analyses did not change the associations (standardized total costs: aGMR = 1.22 (1.07; 1.39), *p* = 0.003; and direct costs: aGMR = 1.27 (1.05; 1.53), *p* = 0.013). Excluding children with asthmatic symptoms in the first three years of life also did not change the associations: standardized total costs: aGMR = 1.28 (1.12; 1.47), *p* < 0.001; and direct costs: aGMR = 1.37 (1.12; 1.69), *p* = 0.002. Stratifying the analyses by the first, second or third year of life did not alter the results (Figure 3). There were no differences in cost with respect to whether the cesarean delivery was planned or acute: standardized total costs: aGMR = 1.07 (0.84; 1.37), *p* = 0.57; and direct costs: aGMR = 1.02 (0.69; 1.52), *p* = 0.92). Appendix A shows the incidences of diagnoses from Figure 1 stratified by delivery mode and Appendix A shows the distribution of costs from Figure 2 stratified by delivery mode.

## 4. Discussion

### 4.1. Primary Findings

We found that the median observed total cost of illness in the first three years of life among 559 unselected, population-based Danish children was EUR 10,932 and the median number of days absent from daycare was 24. Birth by cesarean delivery was associated with 30–45% higher costs and a 17% higher number of days absent from daycare due to illness. Cesarean delivery was significantly associated with all measures of health costs, both direct and indirect, and the association persisted throughout all three study years. Male sex, low GA and low birth weight were also associated with higher costs of child illness, whereas fish-oil supplementation during pregnancy was associated with 12% lower health-related costs in the offspring during the first three years of life. These findings may be useful for developing strategies to avoid or reduce costs of childhood illness.

### 4.2. Strengths and Limitations

A major strength of the study is the longitudinal, prospective clinical surveillance of a mother–child cohort at the COPSAC clinical research unit. The study is a single-center study with assessments by experienced study-physicians, examining the cohort children and obtaining the clinical history based on standard operating procedures supported by daily diary cards. We only included children with at least 90% of days actively observed. This method assures a high sensitivity, consistency in procedures, uniform case definitions, secure data capture methods, and reduced risk of misclassification.

The data source for assessing costs is unique due to the possibility of linking data from national registries to each cohort participant via the Central Person Registration number, which is assigned to all persons with a permanent residence in Denmark. The Danish National Registries contain information on all the population’s contacts with the Danish healthcare system due to the reimbursement system, where the providers are not reimbursed without a correct registration, which strengthens the credibility of the Danish National Registries. The private sector in Denmark is neglectable, particularly regarding children, with no inpatient facilities, thus equalizing the gap in healthcare differences within the population. Still, the costs in the register are average numbers and not calculated for individual hospitalizations; however, we do not expect the numbers to be systematically over- or underestimated.

It is a strength that we were able to consider the burden of parental absenteeism as a part of costs related to childhood illness, which are rarely included in cost analyses. We acknowledge that there might be additional intangible costs, such as reduced quality of life, and secondary transmissions of illness from child to parent, which we were not able to include, and the true cost of childhood illness is probably still underestimated.

The generalizability of our findings is limited by an unintended recruitment bias towards families with atopic illness, conferring a higher risk of those illnesses in their children. In fact, half of the parents were diagnosed with an atopic illness, which is higher than expected in the population [28]. Still, considering the high prevalence of asthma among adults of reproductive age in industrialized countries [29], our results are relevant to a substantial proportion of the population. We observed a high rate of hospital admissions among children (56%), but this is in line with recent studies in a similar Danish population [30]. The generalizability can also be questioned by the relatively good health of our families with low rates of risk factors for poor health such as prematurity, parental smoking, overweight etc. It is therefore possible that we underestimate the level of healthcare utilization. However, families with better health risk profiles could potentially have other doctor seeking behaviors and costs could be higher in this subgroup.

The level of household income in our study is marginally higher as compared to the general population [12], which could lead to an overestimation of the indirect costs of illness. With that in mind, we also estimated the standardized total costs of illness using a fixed daily salary based on average incomes in both Denmark, UK and USA.

Finally, the association between fish-oil supplementation during pregnancy and total costs of child illness is specifically strengthened by the RCT study design.

### 4.3. Interpretation

We found that fish-oil supplementation from pregnancy week 24 to one week postpartum reduced the costs of illness in early childhood by 12%. This suggests fish-oil supplementation for pregnant women as a simple way to reduce the health-related costs of early childhood illnesses. The estimated health expenditure benefit from fish-oil supplementation of 12% of the median observed total cost corresponds to EUR 1400 per child. When we take into account that 61,397 children were born in Denmark in 2017, this reduction of cost amounts to EUR 85,400,000 yearly. Even though full compliance with such future supplementation strategy cannot be expected from the entire population of pregnant women, this would still have the capacity to significantly reduce the health-related costs of childhood illness worldwide.

Children born by cesarean delivery had 30–45% higher costs of illness than children born by vaginal delivery without any differences between planned or acute cesarean delivery. The association between cesarean delivery and increased costs was not only caused by a higher risk of hospitalization perinatally since both total and direct costs of illness without the costs of hospitalizations were significantly associated to the mode of delivery, which persisted during all three years. The association between cesarean delivery and the costs of illness was not mediated by an increased number of infants born prematurely with subsequent increased morbidity, since the associations were still significant when they were adjusted for GA and even when we excluded children born before GA 37 weeks.

Cesarean delivery is a known risk factor for childhood asthma (8), and it is therefore plausible that an increased risk of asthma was driving the association between mode of delivery of costs of illness. Other studies have also shown that cesarean delivery is associated with a higher burden of respiratory illness in childhood [31,32], which fit well with our results of higher costs. However, the association remained significant when excluding children experiencing asthma-like symptoms at age 0–3 years. The incidence rate of cesarean sections is increasing in Denmark, as in many other countries [33], but there are considerable regional differences [34] depending on local obstetrical practices and maternal preferences. The World Health Organization (WHO) has estimated that caesarean section rates above 10% are not correlated with reduced maternal and newborn mortality rates at population level [35], suggesting that it might be possible to lower the rates without health consequences. Our findings stress a potential adverse health effect to the child as well as to society through increasing cost of illness when choosing a cesarean delivery. This is particularly an issue when the procedure is not absolutely medically indicated. Such knowledge on the long-term effects of the procedure may support the clinician, the families and the general healthcare providers in their decision making.

It should be noted that our study identifies an association between cesarean delivery and future health costs with no speculations on the exact mechanisms. Our findings could reflect inherent aspects of susceptibility to illness that also increase the risk of cesarean delivery, and that children born by cesarean delivery constitute a preselected group of more vulnerable children. Future clinical studies can hopefully elucidate these associations.

The fact that low birth weight and premature birth are also associated with healthcare costs throughout early childhood, independently of delivery mode, is consistent with previous studies demonstrating that preterm infants have a significantly higher financial burden throughout their childhood [36,37]. The remarkable effect of prenatal fish-oil supplementation points towards the pivotal importance of the pregnancy and neonatal period, and highlights the importance of optimizing prenatal care, even in high-income settings.

This study provides an overview of the costs of illness in Danish children in general, adding to the current knowledge on the economic burden of childhood illness in affluent countries. The issue about parental absenteeism is regularly a part of the public debate on work–life balance, affecting not only the children and their parents but also the employers, and means by which to alleviate this burden are highly relevant.

## 5. Conclusions

Common childhood illnesses are associated with significant health-related costs to society. These costs can potentially be reduced by targeting the pregnancy and perinatal period, including supplementing maternal diet with fish-oil, reducing the rate of cesarean delivery by maternal request, and by preventing preterm birth and low birth weight.

## Figures and Tables

**Figure 1 children-08-00173-f001:**
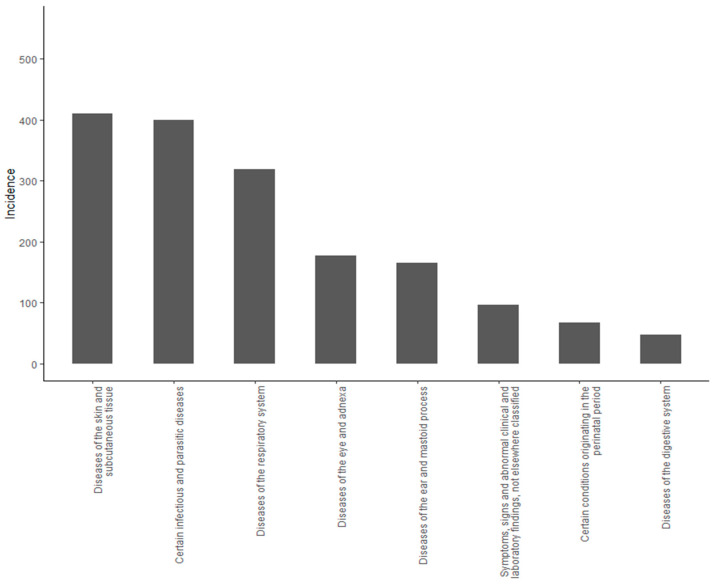
Incidence of disease. The incidence of the eight most common disease by ICD10-group among the 559 participants during the first three years of life.

**Figure 2 children-08-00173-f002:**
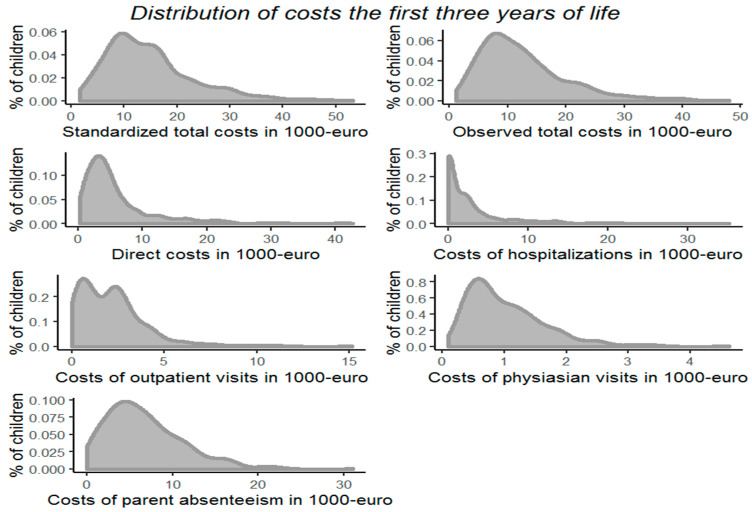
Distributions of costs. Distributions of costs during the first three years of life.

**Figure 3 children-08-00173-f003:**
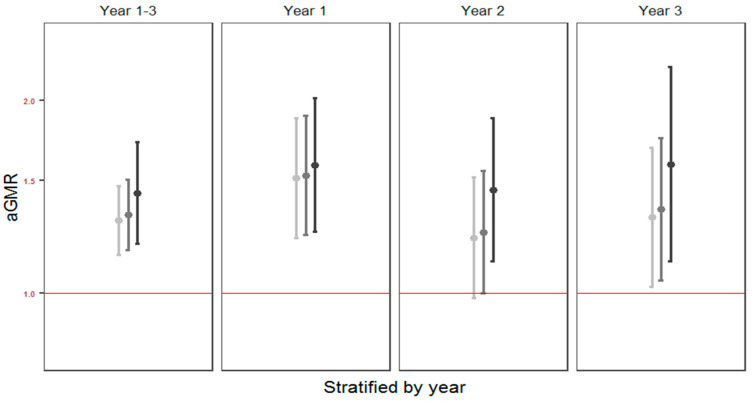
Costs by mode of delivery. The associations between cesarean delivery and costs adjusted for fish-oil supplementation during pregnancy. Adjusted geometric mean ratios of the associations between cesarean delivery and standardized total costs (light grey), observed total costs (middle grey) and direct costs (dark grey) with corresponding 95% confidence intervals stratified by year of life.

**Table 1 children-08-00173-t001:** Baseline characteristics of the study population.

	Study Population
Fish-oil supplement, *n* (%)	282 (50.5)
High-dose vitamin D supplement, *n* (%)	230 (41.1)
Maternal smoking, *n* (%)	16 (2.9)
Maternal age (years), mean ± sd	32.30 ± 4.12
Maternal atopic disease, *n* (%)	312 (55.9)
Maternal prepregnancy BMI, mean ± sd	24.56 ± 4.43
Preeclampsia, *n* (%)	29 (5.2)
Cesarean delivery, *n* (%)	110 (19.7)
Male, *n* (%)	282 (50.4)
Premature, *n* (%)	21 (3.8)
GA (weeks), mean ± sd	39.87 ± 1.71
Birth weight (100g), mean ± sd	35.26 ± 5.52
Season of birth, n (%)	
Summer	114 (20.4)
Autumn	120 (21.5)
Winter	171 (30.6)
Spring	154 (27.5)
Older siblings, *n* (%)	313 (56.0)
Cat at birth, *n* (%)	111 (19.9)
Dog at birth, *n* (%)	97 (17.4)
Breastfed (months), mean ± sd	3.42 ± 1.93
Social circumstances, mean ± sd	0.02 ± 0.95
Day care (years), mean ± sd	0.90 ± 0.24
Type of daycare (nursery), *n* (%)	352 (66.0)

**Table 2 children-08-00173-t002:** Associations between environmental factors, costs and days absent from daycare during the first three years of life (adjusted for fish-oil supplementation). Significant associations in bold.

	Standardized Total Costs		Observed Total Costs		Direct Costs		Days Home from Daycare	
Environmental Factor	aGMR (95% CI)	Interpretation	aGMR (95% CI)	Interpretation	aGMR (95% CI)	Interpretation	aOR (95% CI)	Interpretation
	(*p*-Value)	(Euro)	(*p*-Value)	(Euro)	(*p*-Value)	(Euro)	(*p*-Value)	(Days)
High-dose vitamin D suppl.	1.05 (0.95; 1.17) 0.32	708 (−660; 2223)	1.07 (0.96; 1.19) 0.20	733 (−378; 1966)	1.04 (0.89; 1.21) 0.62	158 (−439; 853)	0.99 (0.88; 1.12) 0.90	−0.2 (−3.0; 3.0)
Maternal smoking	1.19 (0.88; 1.61) 0.25	2602 (−1555;8210)	1.12 (0.83; 1.53) 0.46	1304 (−1819; 5557)	**1.60 (1.02; 2.5) 0.039**	2450 (101; 6117)	1.10 (0.79; 1.57) 0.60	2.05 (−5.5; 14.8)
Maternal age	1.00 (0.99; 1.02) 0.43	66 (−97; 231)	**1.02 (1.01; 1.03) 0.004**	195 (64; 329)	1.02 (1.00; 1.03) 0.08	66 (−8; 142)	1.00 (0.98; 1.01) 0.78	−0.1 (−0.4; 0.3)
Maternal atopic disease	1.10 (1.00; 1.22) 0.06	1390 (−31; 2962)	1.07 (0.97; 1.19) 0.19	758 (−347; 1984)	**1.22 (1.05; 1.42) 0.009**	899 (206; 1703)	1.05 (0.93; 1.18) 0.42	1.03 (−1.7; 4.6)
Maternal pre pregnancy BMI	1.00 (0.99; 1.01) 0.60	41 (−111; 196)	1.00 (0.99; 1.01) 0.85	11 (−110; 135)	1.01 (1.00; 1.03) 0.13	53 (−16; 124)	1.00 (0.99; 1.01) 0.87	0.0 (−0.4; 0.3)
Preeclampsia	**1.29 (1.03; 1.61) 0.03**	3841 (360; 8199)	1.25 (0.99; 1.58) 0.06	2618 (−85; 6025)	1.12 (0.80; 1.57) 0.51	484 (−821; 2310)	1.29 (1.01; 1.68) 0.05	7.05 (0.1; 17.6)
Cesarean delivery	**1.30 (1.15; 1.47) <0.0001**	4061 (2021; 6372)	**1.33 (1.17; 1.51) <0.0001**	3424 (1764; 5310)	**1.45 (1.20; 1.74) 0.0001**	1822 (824; 3022)	**1.17 (1.02; 1.35) 0.029**	4.05 (0.5; 9.2)
Male	**1.16 (1.05; 1.28) 0.004**	2105 (631; 3733)	**1.18 (1.06; 1.30) 0.0019**	1852 (655; 3178)	**1.17 (1.01; 1.36) 0.0371**	703 (40; 1471)	1.03 (0.92; 1.16) 0.58	0.09 (−2.0; 4.1)
GA (weeks)	**0.93 (0.91; 0.96) <0.0001**	−901 (−1259; −534)	**0.92 (0.90; 0.95) <0.0001**	−789 (−1071; −499)	**0.84 (0.81; 0.88) <0.0001**	−645 (−785; −499)	1.03 (0.99; 1.06) 0.14	0.07 (−0.2; 1.6)
Premature	**2.00 (1.55; 2.59) <0.0001**	13492 (7392; 21378)	**2.21 (1.70; 2.87) <0.0001**	12606 (7278; 19536)	**4.08 (2.81; 5.94) <0.0001**	12600 (7391; 20175)	0.86 (0.64; 1.17) 0.31	−3.7 (−9.4; 4.5)
Birth weight (100g)	**0.98 (0.97; 0.99) 0.0003**	−225 (−343; −105)	**0.98 (0.97; 0.99) 0.0002**	−186 (−281; −90)	**0.96 (0.95; 0.98) <0.0001**	−144 (−196; −92)	1.00 (0.99; 1.01) 0.69	−0.1 (−0.3; 0.2)
Season of birth (ref: summer)								
Autumn	1.07 (0.92; 1.25) 0.37	983 (−1087; 3398)	1.10 (0.94; 1.29) 0.25	1032 (−658; 3014)	1.09 (0.86; 1.37) 0.48	351 (−564; 1504)	1.05 (0.88; 1.26) 0.56	1.40 (−3.0; 6.7)
Winter	0.93 (0.81; 1.08) 0.35	−895 (−2566; 1033)	0.93 (0.81; 1.08) 0.36	−692 (−2025; 852)	0.95 (0.77; 1.18) 0.66	−192 (−939; 734)	0.94 (0.80; 1.11) 0.50	−1.40 (−5.2; 2.9)
Spring	0.98 (0.85; 1.14) 0.81	−243 (−2036; 1831)	0.99 (0.85; 1.14) 0.84	−156 (−1591; 1511)	0.98 (0.78; 1.21) 0.82	101 (−881; 869)	1.00 (0.85; 1.18) 0.99	0.0 (−4.0; 4.8)
Older siblings at birth	1.08 (0.97; 1.19) 0.16	1015 (−372; 2549)	**1.11 (1.00; 1.23) 0.0496**	1146 (4; 2412)	1.06 (0.92; 1.24) 0.42	262 (−345; 967)	**1.22 (1.04; 1.44) 0.015**	1.6 (−1.4; 5.0)
Cat at birth	0.95 (0.84; 1.08) 0.45	−639 (−2148; 1073)	0.91 (0.80; 1.03) 0.14	−969 (−2115; 336)	0.95 (0.79; 1.15) 0.63	−184 (−849; 618)	1.02 (0.83; 1.26) 0.83	0.40 (−3.1; 4.5)
Dog at birth	0.95 (0.83; 1.08) 0.43	−701 (−2284; 1107)	0.92 (0.80; 1.05) 0.21	−879 (−2100; 521)	1.21 (0.99; 1.47) 0.06	843 (−39; 1917)	0.82 (0.66; 1.03) 0.08	−4.2 (−7.2; −0.5)
Social circumstances	0.99 (0.94; 1.05) 0.85	−68 (−753; 654)	NA	NA	0.97 (0.90; 1.05) 0.43	−127 (−425; 196)	1.00 (0.92; 1.09) 0.97	−1.0 (−1.6; 1.5)
Solely breastfed (mths)	1.00 (0.98; 1.03) 0.89	25 (−320; 379)	1.00 (0.98; 1.03) 0.75	45 (−231; 329)	0.97 (0.93; 1.01) 0.10	−131 (−280; 25)	1.00 (0.96; 1.04) 0.97	0.3 (−0.5; 1.0)
Introduction to daycare (mht)	**0.97 (0.95; 0.99) 0.0009**	−394 (−621; −164)	**0.97 (0.96; 0.99) 0.004**	−274 (−456; −89)	1.00 (0.97; 1.02) 0.90	−7 (−113; 102)	**0.96 (0.93; 1.00) 0.009**	−1.3 (−1.9; −0.7)
Type of daycare (nursery)	**1.23 (1.11; 1.37) 0.0001**	3101 (1440; 4948)	**1.27 (1.13; 1.41) <0.0001**	2773 (1409; 4293)	0.94 (0.80; 1.11) 0.48	−232 (−809; 446)	**1.33 (1.12; 1.58) 0.001**	9.0 (5.1; 13.3)

## Data Availability

All materials, computer code and protocols will be made available to readers. However, the data included in the study are classified as sensitive, personally identifiable data. Participant-level personally identifiable data are protected under the Danish Data Protection Act and European Regulation 2016/679 of the European Parliament and of the Council (GDPR) that prohibit distribution even in pseudoanonymized form, and these data therefore cannot be made publicly available, but can potentially be made available under a data transfer agreement as a collaboration effort.

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
