# Peer review of "Cost of Illness in Young Children: A Prospective Birth Cohort Study"

_children, 2021, doi:10.3390/children8030173_

Round 1
Reviewer 1 Report
The manuscript provides a significant topic. The reviewer is struggling to identify what do authors mean by saying common childhood illness. It would be helpful to name them. The study is a limited attempt towards the topic. I have marked few comments in the reviewed file. Due to the PDF format of the reviewing file, It's not convenient to mark language and grammar corrections.

Reviewer 2 Report
Thank you for sharing these interesting findings in a well designed, well constructed study with great presentation of the data and a thoughtful discussion on the impact and limitations of the trial. I have some feedback below and suggestions to consider:
- One of the things I asked myself as I read your manuscript was the differentiation between outpatient and inpatient procedures and cost. You very nicely broke down the burden of childhood disease, but was not distributed in terms of outpatient vs inpatient (and ED visits). Do you have any descriptions of this data?
- I know this data was collected as part of the original study, but I don't see the relevance of the pet data.
- I thought that your conclusion section was particularly strong.
- You addressed many key points and limitations to your data and conclusions, such as additional costs to the family beyond just parental absenteeism.
- Mentioned the distinction between causation and association between C-Section and costs/greater health care needs.
- Could make a stronger mention of the limited generalizability of the study. Your population was relatively healthy, despite the high rates of atopic illness. For example, the population had low rates of prematurity, low rates of smoking, normal BMI, low rates of PEC, etc. I would address this in your discussion as well. Also, do you have other demographic information like race, ethnicity, maternal education?
- Would have also liked to have seen a slightly stronger mention of preterm birth and low birth weight in the discussion, as these many be preventable with stronger obstetrical interventions and research. You mention C-Section very thoroughly but would have liked a little more mention of preterm birth, as there is a good deal of data that preterm infants confer a significant amount of financial burden throughout their lifetimes.
Reviewer 3 Report
The manuscript is written according standards.
